# Investigation of Potential of GNSS-R Polarization: Theoretical Simulations

**Xuerui Wu** [1,2,†] [iD]**, Xiaoyong Du** [3,†]**, Feng Yan** [4,*]**, Weihua Bai** [5] **and Shaohui Song** [6]

1    School of Resources, Environment and Architectural Engineering, Chifeng University, Chifeng 024000, China; xrwu@shao.ac.cn
2    Shanghai Astronomical Observatory, Chinese Academy of Sciences, Shanghai 200030, China
3    Beijing Institute of Applied Meteorology, Beijing 100029, China; duxiaoyong@mail.hzau.edu.cn
4    Shenzhen Aerospace Dongfanghong Satellite Ltd., Shenzhen 518054, China
5    National Space Science Centers, Chinese Academic of Sciences, Beijing 100190, China; baiweihua@nssc.ac.cn
6    School of Remote Sensing and Geomatics Engineering, Nanjing University of Information Science and Technology, Nanjing 210044, China; 20201248063@nuist.edu.cn
*    Correspondence: yanfeng@szhtdfh.com
†    These authors contributed equally to this work.

**Abstract:** Global navigation satellite system (GNSS) reflectometry (GNSS-R) developed into a promising remote sensing technique. However, few previous related studies considered the potential of its polarization. Owing to lack of sufficient in situ measurement data to support comprehensive investigation of GNSS-R polarization, this study used theoretical models and reference to our previous work to explore this topic. The commonly used microwave scattering models are employed to get the bare soil or vegetation scattering properties of GNSS-R configurations, i.e., the random surface scattering model and the first-order radiative transfer equation were improved and then employed to obtain the scattering properties of both bare soil and vegetation. Since the final output of the space-borne GNSS-R missions is a delay Doppler map (DDM), a spaceborne (DDM) simulator, oriented for the Chinese FengYun-3E (FY-3E) GNSS-R payload, was utilized to obtain the final output at different polarizations. Using the developed models (such as the bare soil and vegetation scattering models), corresponding polarization simulations were performed. That is to say, not only the commonly used LR (left hand circular polarizations (LHCP) received and the right hand circular polarizations (RHCP) received) can be presented, but also the scattering properties at RR, VR, and HR (the transmitted signals are RHCP, while the received polarizations are RHCP, vertical (V) and horizontal (H) polarizations, respectively) can be predicted by our developed models. Results reveal obvious polarization differences for the bistatic scattering and DDM. Therefore, the use of GNSS-R polarization information has potential to provide competitive and fruitful results in the future detection of land surface geophysical parameters.

**Keywords:** GNSS-reflectometry; DDM; polarization; soil; vegetation

## 1. Introduction

Global navigation satellite system (GNSS) is a general term used for various navigation satellite systems. Following completion of China's Beidou global satellite network, there will be more than 150 in-orbit satellites comprising various GNSSs. The direct signals of GNSSs that are reflected from the ground surface can be used in a new type of remote sensing technique, i.e., GNSS reflectometry (GNSS-R) [1–3]. Its applications extend from the ocean surface to land surface, such as soil moisture, vegetation biomass, wetland monitoring, and soil freeze and thaw detection [4–7].

The transmitters and corresponding GNSS-R receivers form a typical bistatic radar employ electromagnetic waves (typically L-band waves) to detect the surface of the earth.

It should be noted that polarization, which is one of the most important features of electromagnetic waves, can be defined as the orientation and amplitude of the electric field strength change over time. However, with regard to GNSS-R, the polarization properties remain an open issue [8].

In the earliest soil moisture experiment 02/03 (SMEX02/03), the delay map receiver was used to obtain the surface-reflected signals. It was considered that the surface-reflected signals had left-hand circular polarization (LHCP) and thus the LHCP antenna was used to collect the surface-reflected signals, while the right-hand circular polarization (RHCP) antenna was used to obtain the direct signals [9]. However, in the following BAO tower experiment, an improved delay map receiver was used, and four antennas (RHCP, LHCP, horizontal (H), and vertical (V)) were employed. The intention was to use the polarization information to remove the effects of surface roughness on the final soil moisture retrievals [9]. It was believed that the simplifying assumptions of the theoretical model could explain the differences between the models and the in situ measurements [10]. The researchers adopted improved geodetic-quality receivers, i.e., the soil moisture interference pattern GNSS observations at L-band reflectometer (SMIGOL) and the dual-polarization SMIGOL (PSMIGOL), to perform their airborne experiments, and used the linear polarization (V and H) antennas to obtain the surface-reflected signals. Using the interferometric pattern signals collected by the receivers, the authors employed notch positions and numbers to retrieve soil moisture and vegetation water content [11–13]. The SAM receiver was used in the land monitoring with navigation signals (LEiMON) experiment, together with RHCP and LHCP antennas. The relationships between the target physical parameters (e.g., surface roughness, soil moisture content, and vegetation water content) and the GNSS-R signal revealed a strong correlation between the LR signal, RR signal, and vegetation water content (correlation coefficient: 0.8) [14].

As for spaceborne GNSS-R missions, the UK-DMC satellite launched in 2004 was equipped with the first spaceborne GPS-R payload. It was initially designed to study ocean surface roughness and to estimate sea surface wind speed, but later studies found that signals from the land surface could also be collected [15]. On 8 July 2014, the TechDemoSat-1 (TDS-1) satellite was successfully launched. It is equipped with a GNSS reflected-signal receiver, i.e., the space GNSS receiver remote sensing instrument (SGR-ReSI). TDS-1 provides a large amount of GNSS reflection data from the land surface [16]. The Cyclone Global Navigation Satellite System (CYGNSS) was launched in 2016. The main scientific goal of CYGNSS is detection of hurricane and ocean parameters [7], but recent studies demonstrated the advantages of CYGNSS with regard to the estimation of typical hydrological essential climate variables [4–6,17,18]. It is worth noting that the UK-DMC, TDS-1, and CYGNSS all operate on the assumption that signals reflected from the surface have LHCP, which is reflected in the polarization of the receiver antenna. Following hardware failure of the SMAP satellite, it was converted to GNSS-R mode (i.e., SMAP-R), where H polarization and V polarization antennas are employed to receive the reflected signals [19,20]. The European Space Agency plans to launch the HydroGNSS in 2024, the scientific objective of which is the detection of essential climate variables using the dual polarization of the corresponding receiver antennas [21].

In summary, regardless of whether ground-based, airborne, or spaceborne GNSS-R is used, the polarization properties are not definitive. However, given the importance of using GNSS-R polarization in remote sensing of the land surface, corresponding research should be undertaken.

Owing to the lack of targeted experiments, research based on the theoretical mechanism model is especially important. Theoretical study can assist in subsequent experimental design, data analysis, terrestrial receiver design, and quantitative inversion of parameters [22]. Therefore, based on our previous work related to GNSS-R polarization, this study employed theoretical models to undertake simulation analysis to promote the scientific development and engineering design of GNSS-R polarization technology.

Our developed models are based on the previous backscattering microwave scattering models and they are presented in Section 2. The theoretical simulations are described and analyzed in Section 3. The derived conclusions are presented in Section 4.

## 2. Model Development

For bare soil and vegetation, the random surface scattering model and first-order radiative transfer (RT) model are employed to calculate geophysical parameters [23,24]. On 5 July 2021, the Chinese meteorological satellite FY-3E was successfully launched. It carries 11 payloads, one of which is the GNSS occultation and GNSS-R combined payload. In conjunction with the FY-3E GNSS-R mission, our land surface GNSS-reflection simulator (LAGRS) is also employed to obtain the polarization characteristics of surface objects [25]. However, it should be noted that owing to the characteristics of GNSS-R, the scattering models suitable for conventional microwave remote sensing must be improved. This section introduces methods for such improvements, together with details of the developed LAGRS.

### 2.1. Random Surface Scattering Models

In the microwave region, the scattering models commonly used for a natural surface are (1) the geometrical optics (GO) model, (2) the physical optics (PO) model, (3) the small perturbation model (SPM), and (4) the integral equation model (IEM) or advanced integral equation model (AIEM). These models are suitable for natural surfaces with different surface roughness. Broadly, the GO model is best suited for very rough surfaces, the PO model is more suitable for a surface with intermediate scales of roughness, and the SPM is suited for surfaces with a short correlation length, which is the scalar of surface roughness. The ranges of roughness over which these models should be applied are presented in Table 1. The IEM is based on the complex integral model and it is suitable for surfaces with continuous roughness. The AIEM is more complex than the IEM, but it provides comparatively better accuracy [26].

**Table 1.** Application range of the GO model, PO model, and SPM [4].

| Model Name | Valide Conditions | | | Recommended Conditions | |
|---|---|---|---|---|---|
| GO | $s \geq \lambda/3$ | $l_s \geq \lambda$ | $l_s^2 > 2.76s\lambda$ | $0.4 \leq m \leq 0.7$ | |
| PO | $0.05\lambda \leq s \leq 0.15\lambda$ | $l_s \geq \lambda$ | $m \leq 0.25$ | $l = \lambda/6, l_s \geq 6l$ | $0.05\lambda \leq s \leq 0.15\lambda$ |
| SPM | $s \leq 0.05\lambda$ | $m \leq 0.3$ | $l_s \leq 0.5\lambda$ | $l_s \leq 0.25\lambda$ | $s \leq 0.05\lambda$ |

### 2.2. First Order Radiative Transfer Model

The vegetation model applied to a GNSS-R bistatic radar can be divided into two types, one of which is an approximate field method for weak scattering media. For vegetation, a strong scatterer is discrete, and the dielectric constants of such scatterers are much larger than those of air. Therefore, it is more suitable to use radiation transfer theory to solve the scattering problem. At the same time, the vegetation model can be divided into a continuous model and a discrete model. The continuous vegetation model treats the vegetation layer as a continuous fluctuating medium. The discrete vegetation model considers that the vegetation layer can be treated as a single scatter with different size, shape, and spatial distribution probability. In this study, when analyzing vegetation, a double layer noncoherent scattering model for high crown layers was employed, which divides the vegetation layers above the surface into the crown and the trunk layer [24]. The model is based on the Stokes matrix of a single scatterer. It obtains the corresponding phase matrix and finally derives the phase matrix of the respective scatters.

### 2.3. Coordinate Change

To calculate the scattering properties of different geophysical parameters using GNSS-R requires consideration of bistatic scattering. Therefore, the forward scattering alignment

(FSA) coordinate system should be employed (Figure 1) because FSA mainly applies to the bistatic scattering of nonuniform media [26–30].

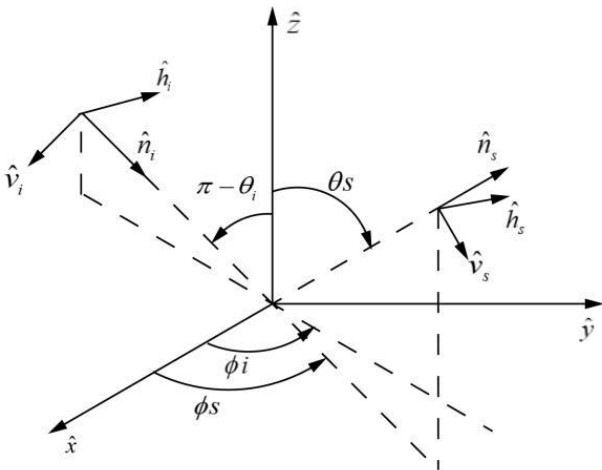

**Figure 1.** Forward scatter alignment.

Vertical and horizontal vectors in FSA are defined along the direction of wave propagation. The coordinate system is consistent with the standard spherical coordinate system. The definitions of an incident wave and a scattering wave are as follows:

$$n_i = \hat{x}sin\theta_i cos\phi_i + \hat{y}sin\theta_i sin\phi_i + \hat{z}cos\theta_i \tag{1}$$

$$\hat{h}_i = -\hat{x}sin\phi_i + ycos\phi_i \tag{2}$$

$$\hat{v}_i = \hat{h}_i \times \hat{n}_i = \hat{x}cos\theta_i cos\phi_i + \hat{y}cos\theta_i sin\phi_i - \hat{z}sin\theta_i \tag{3}$$

$$\hat{n}_s = \hat{x}sin\theta_s cos\phi_s + \hat{y}sin\theta_s sin\phi_s + \hat{z}cos\theta_s \tag{4}$$

$$\hat{h}_s = -\hat{x}sin\phi_s + \hat{y}cos\phi_s \tag{5}$$

$$\hat{v}_s = \hat{h}_s \times \hat{n}_s = \hat{x}cos\theta_s cos\phi_s + \hat{y}cos\theta_s sin\phi_s - \hat{z}sin\theta_s \tag{6}$$

### 2.4. Wave Synthesis Technique

Traditional microwave scattering models are suitable for studies involving typical linear polarizations; however, for GNSS-R, the transmitted signals of GNSS constellations are RHCP. To study GNSS-R polarization, we need to modify the model to make it suitable for circular polarizations and we need to obtain various polarization properties. Therefore, the wave synthesis technique is employed in model development. The polarization state of the wave may also be expressed in terms of the Stokes vector, in which the orientation $\psi$ and ellipticity angles $\chi$ are sufficient for complete specification of the polarization. The bistatic scattering cross section for any combination of transmit and receive polarizations can be written as follows:

$$\sigma_{rt}(\psi_r, \chi_r, \psi_t, \chi_t) = 4\pi \widetilde{Y}_m^r M_m Y_m^t \tag{7}$$

where $\widetilde{Y}_m^r$ and $Y_m^t$ are the normalized Stokes vectors characterizing the transmitter and receiver polarizations. $M_m$ is the modified Mueller matrix.

### 2.5. Spaceborne DDM Model

Currently, there is no spaceborne GNSS-R payload that is oriented to GNSS-R polarization. However, spaceborne models can be adopted for analysis. In this section, we describe the adoption of the extended version (full polarization) of LAGRS, which is a simulator for the FY-3E GNSS-R payload [25,31].

LAGRS is an integral form of bistatic radar that is based on the ocean surface GPS scattering model During our development of LAGRS, we extended it for application to bare soil and vegetation. The main modifications involved converting the integration area, observation geometry, and target microwave scattering models. Using this model, we can describe the GPS and Beidou reflection power for a different time delay $\hat{\tau}$ and Doppler frequency shift $\hat{f}$:

$$Y_s\left(\hat{\tau}, \hat{f}\right) = \frac{T_I^2 P_T \lambda^2}{(4\pi)^3} \iint_A \frac{G_T \sigma^0 G_R}{R_R^2 R_T^2} \Lambda^2(\hat{\tau} - \tau) sinc^2\left(\hat{f} - f\right) dA \tag{8}$$

where $Y_s$ is the GPS or Beidou reflection power, $P_T$ is the transmitted power of the GNSS constellation, and $G_T$ and $G_R$ are the antenna gain of the transmitter and receiver, respectively. $R_R$ is the distance from the receiver to the ground scattering point, $R_T$ is the distance from the transmitter to the ground scattering point, $\lambda$ is the electromagnetic wavelength, $T_I$ is the integration time, $\sigma^0$ is the bistatic scattering cross section, which can be calculated using the models described in Sections 3.1 and 3.2, $\Lambda(\hat{\tau} - \tau)$ is the GPS correlation function (triangle function), $\hat{\tau}$ and $\tau$ are the time delays of the duplicated and incident signals, respectively, $sinc^2\left(\hat{f} - f\right)$ is the doppler filter function, $\hat{f}$ and $f$ are the frequency of the duplicated and incident frequencies, respectively, $A$ is the effective scattering area, which is approximative to the glistening zone, and $dA$ is the integration area in zone $A$. Figure 2 presents a flowchart of the LAGRS model. The bistatic scattering model is obtained from the previously mentioned random surface scattering model and the first-order RT model. The velocity and position of the transmitter and receiver can be used to obtain the geometry module. By combining the two modules, we can determine the final delay Doppler map (DDM) of the corresponding geophysical parameters.

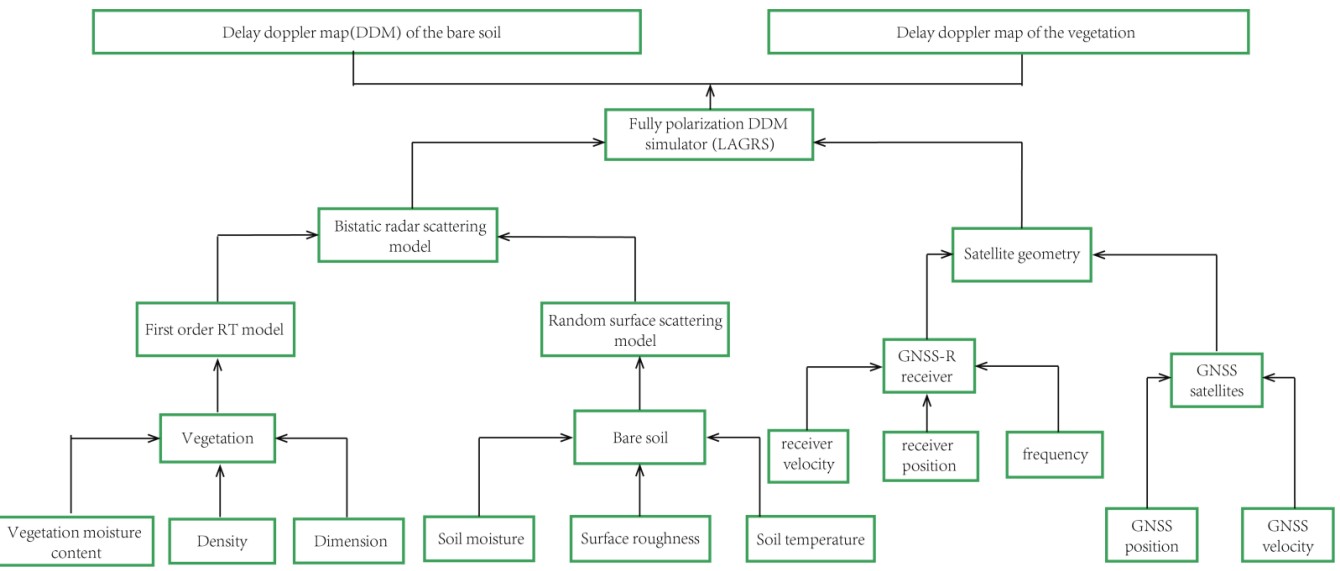

**Figure 2.** Flowchart of the LAGRS model.

## 3. Theoretical Simulations

Using the developed models presented in Section 2, we describe the corresponding scattering properties and simulated spaceborne DDM output at various polarizations.

### 3.1. Bare Soil DDM Output at Various Polarizations

Using the model described in Section 2.1, we present the bare soil simulations at various polarizations [28]. Given that GNSS-R is a type of bistatic radar, the observation geometry will vary considerably and these variations will result in the scattering difference.

Figure 3 illustrates the scattering simulations. The panels of first line to the fifth line represent the scattering properties of the LR, VR, HR+45°, and R−45° R polarizations, respectively, where LR, VR, HR+45°, and R−45° R refer to received polarizations with LHCP, and V, H+45°, and −45° refer to transmitted polarizations with RHCP. The volumetric soil moisture content from the first column to the third column is 0.1, 0.3, and 0.6, respectively. From the *x* axis and the *y* axis, it can be seen that the observation geometry covers almost all possible azimuth and zenith angles. The simulations show that the scattering properties vary considerably and that these variations relate to the observation geometry, polarization, and soil moisture content. However, extraction of soil moisture information from these influencing factors represents important work that should be undertaken urgently.

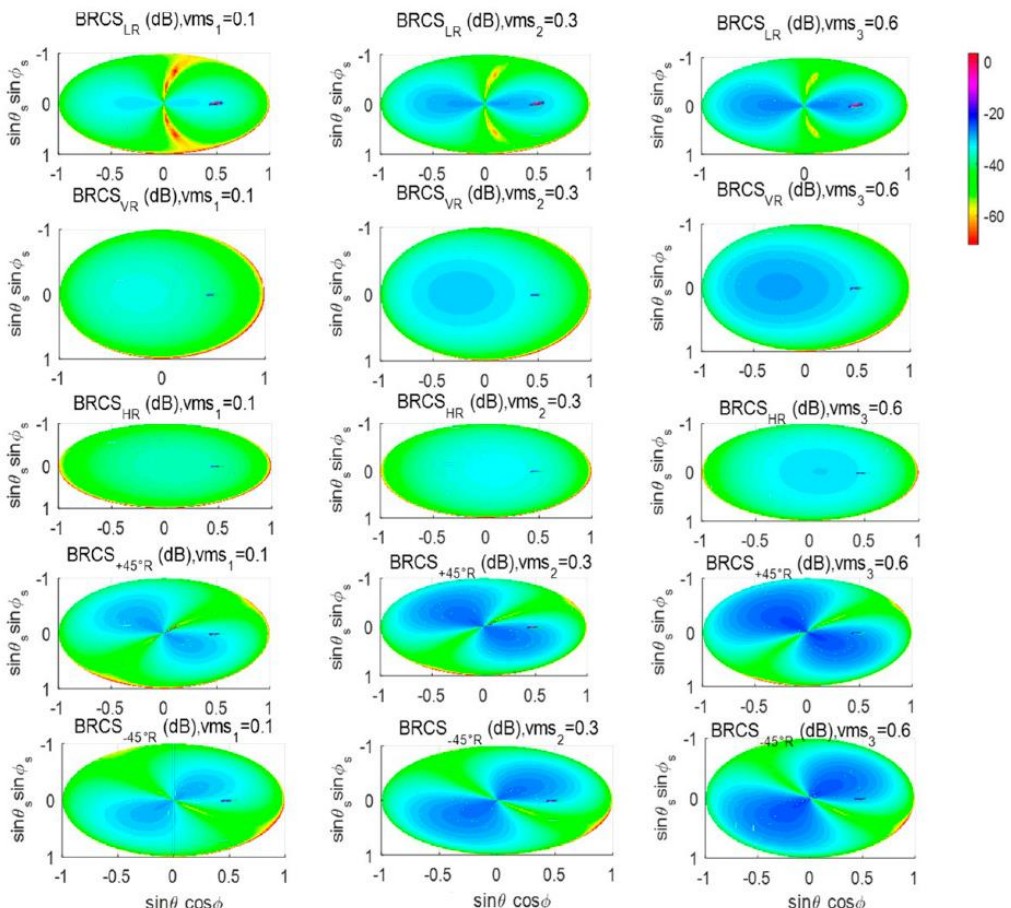

**Figure 3.** Bistatic scattering properties at various polarizations. vms is the volumetric moisture content of the soil. The x-axis and y-axis indicate the different scattering geometries.

To obtain spaceborne information, we employed LAGRS to simulate the final DDM output. During our simulations, the integration areas were set as defined in Table 2, where X_range and Y_range are the spatial integral range in the *x* direction and *y* direction, respectively, and X_interval and Y_range are the spatial integral interval in the *x* direction and *y* direction, respectively.

Our objective was to obtain the polarization effects; therefore, we kept other influencing factors constant. The volumetric soil moisture content was set 0.3.

**Table 2.** Integration area settings.

| Integration Area | |
|---|---|
| X_range | $[-40,40] * 1000$ |
| Y_range | $[-40,40] * 1000$ |
| X_interval | 1000 |
| Y_interval | 1000 |

The spaceborne DDM output realized using the LAGRS model is presented in Figure 4, paper [8] presents more detail information. The DDM output at LR, VR, and HR polarizations is shown in Figure 4, respectively. From the simulations, it can be seen that the scattering properties at the specular point have certain variations, while the difference at the glistening scattering zone varies considerably. The final DDM output is the power integrated from different observation geometries, and the scattering variations at various observation angles are presented in Figure 3. These scattering properties result in the final DDM variations that are presented in Figure 4.

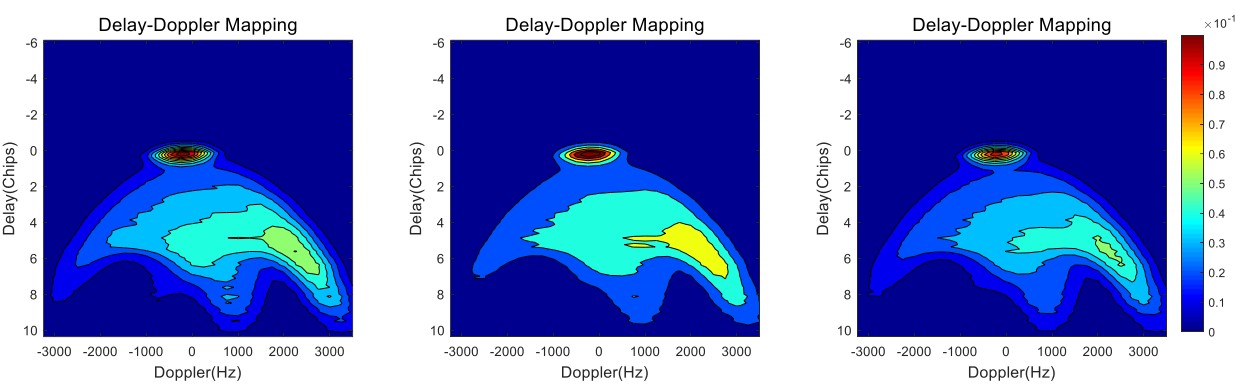

**Figure 4.** DDM variations at LR (**left**), VR (**middle**), and HR (**right**) polarizations.

### 3.2. Vegetation DDM Output at Various Polarizations

In this section, we present the scattering properties of vegetation and the corresponding spaceborne DDM output using the models described in Section 2.2. Because the target geophysical parameters have much larger scattering coefficients at the specular scattering cone, we first present the coherent scattering properties at various polarizations.

It can be seen from the left figure of Figure 5 that the scattering properties vary as the polarization changes. Scattering at RR polarization has the lowest values for lower incidence angles; the values increase as the incidence angle increases, and finally decrease at larger incidence angles. Except for RR and VR polarizations, the scattering properties of the other polarizations present trends that are similar. The scattering properties at VR polarization should be given greater attention because of an apparent decrease in the incidence angle of 70°, a phenomenon attributable to lower scattering at the Brewster angle. The component scattering contribution to the total scattering at LR polarization is presented in the right figure of Figure 5. It can be seen that the total scattering is dominated by the trunk layer, and that the scattering from the crown layer is the lowest. The scattering information from the ground parts (i.e., the direct ground component and the specular ground component) has higher scattering values than the crown layer.

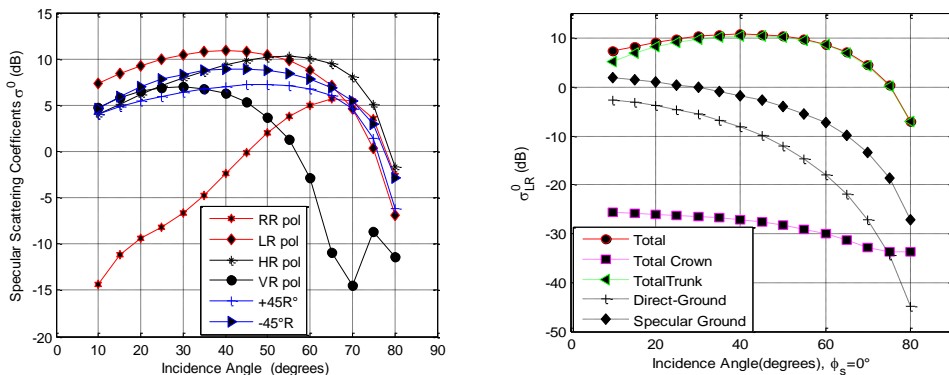

**Figure 5.** The vegetation specular scattering properties. The left figure is the coherent scattering at various polarizations. The right figure is the component scattering contribution to the total scattering.

By employing the LAGRS simulator, the DDM output at different polarizations is realized (Figure 6). Figure 6 (left figure) shows the DDM at LR polarization, while the DDM at VR and HR polarization is shown in Figure 6 (middle figure) and 6 (right figure), respectively. The integration area is listed in Table 2. The vegetation parameters for the simulation in Figure 6 were set as described, and all other observation geometry parameter settings were kept constant. The final DDM output shown in Figure 6 reflects the polarization difference, i.e., the polarization provides different information for the same vegetation given the same observation geometry and integration area.

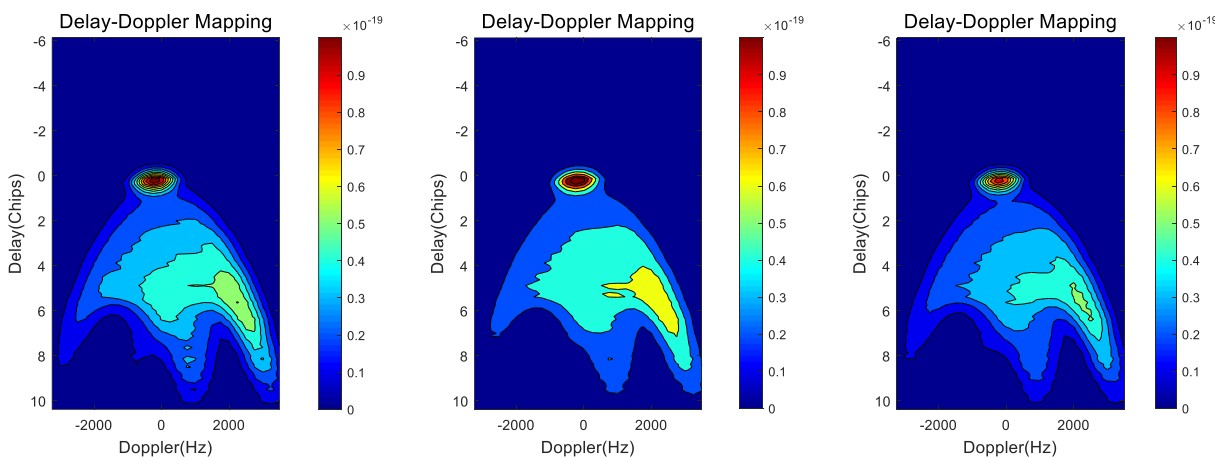

**Figure 6.** Vegetation DDMs at LR polarization (**left**), VR polarization (**middle**), and HR polarization (**right**).

## 4. Conclusions

Establishing how best to fully exploit polarization information for detection of land geophysical parameters is perhaps one of the most important tasks in the promising GNSS-R remote sensing technique. Owing to a lack of sufficient in situ measurement data suitable for such research, we relied on microwave scattering models to further such analysis. To investigate the potential of polarization properties for spaceborne GNSS-R, the random surface scattering model and the first-order RT model were employed, together with the corresponding DDM simulator (i.e., LAGRS) that is oriented for the FY-3E GNSS-R payload. The importance of these theoretical models is that they are the key mechanism tools for analyzing and interpreting satellite observations, simulation/assimilation of satellite data, development of quantitative inversion algorithms of surface parameters, and the design of new sensors. The simulations in this paper revealed obvious differences in the DDM outputs and the scattering variations at different polarizations. They indicate

that polarizations of the GNSS-R remote sensing technique can potentially provide very good information for geophysical parameter retrieval. Therefore, we will conduct further theoretical and experimental work on this topic in the near future.

**Author Contributions:** Conceptualization, methodology, draft preparation and supervision X.W.; supervision and funding acquisition, X.D., F.Y. and W.B.; revision tasks and draft preparation, S.S. All authors have read and agreed to the published version of the manuscript.

**Funding:** This research was funded by the National Natural Science Foundation of China (No. 42061057, and 72004017) and the innovative Teams of Studying Environmental Evolution and Disaster Emergency Management of Chifeng University (cfxykycxtd202006) and Chifeng University, Laboratory of National Land Space Planning and Disaster Emergency Management of Inner Mongolia (CFXYZD202006).

**Acknowledgments:** Special thanks are given to the reviewers for their constructive comments and helpful suggestions.

**Conflicts of Interest:** The authors declare no conflict of interest.

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
