# Peer review of "Investigation of Potential of GNSS-R Polarization: Theoretical Simulations"

_remotesensing, doi:10.3390/rs14153700_

Round 1
Reviewer 1 Report
line 16- which work ? have you cited.
line 16- check spelling of previous (s is ommited- include s)
line-20-= which MODEL ? name it
Line 22-23- English comma is required in this sentence. check English Throughout / by native speaker
line 27 Use here full form of term at its first appearance of GNSS.
Line 30- explain its importance to readers about operational applications- there are several retrieval methods for soil moisture content (SMC), altimetry, and vegetation biomass estimation and use proper citation.
Line 81- spelling of summary - u is missing.
line 82- how this model is different from previous one. what was your previous work. readers dont understand this after reading previous work. please elaborate and mentioned what was the work you are referring to.
provide- merit and demerits of developed model.
Preference of your model with previous models.
Author Response
Dear Reviewer,
Thank you very much for your suggestions. For more information of our replies and revisions , please see the MS attachment.
Best regards,
Xuerui

Reviewer 2 Report
Dear Authors:
While carefully reviewing this article, I saw majority of flaws were related to formatting of the MS. Also the MS is very short and I doubt if the structure fits according to the authors guidelines of this journal. For e.g. the abstract and conclusions were very short. I did not find the sections related to results and discussions. The figures and the related text or labels were also very difficult to read or understand. Also there were grammatical errors at several lines in the sentences. Attached please find the reviewed version of the article with all my comments, suggestions and questions (marked inside the PDF it self). Try to revise your MS accordingly.
Best wishes

Author Response
Dear Reviewer,
Thank you very much for your helpful suggestions, we have revised our paper according to your comments point by point.
For more detail information, please see the MS attachment.
Best regards,
Xuerui Wu

Round 2
Reviewer 1 Report
The authors did a lot of good work on the previous version and incorporated the comments and suggestions. The current revised MS version has been significantly improved in scientific writing and as per guidelines. In my opinion, MS is suitable for publication.
Author Response
请参阅附件。

Reviewer 2 Report
Dear Authors:
Attached please find the reviewed version of the MS. You revised the MS. However, the MS still has some formatting issues.
For example:
The styling used in list of references needs through check and revision
The styling used in conflict of interest section needs revision
Also one of the figure (Figure#3) is not fitting within the page limit.
Note: Everything should be according to the authors guidelines (instructions to authors) for this journal, available at:
https://www.mdpi.com/journal/remotesensing/instructions
You must check everything carefully before resubmitting the next version.
All the best!
